# DoubleAgent: Towards Text-to-SQL on Enterprise Databases

Fares Elkholy*
Technical University of Darmstadt

Timo Eckmann*
Technical University of Darmstadt

Jan-Micha Bodensohn
Technical University of Darmstadt

Mulang' Onando
SAP SE

Carsten Binnig
Technical University of Darmstadt &
hessian.ai & DFKI

## ABSTRACT

Recent progress on Text-to-SQL has almost saturated existing benchmarks such as Spider, BIRD, and Spider 2.0-Snow. These benchmarks, however, rest on the assumption that database schemas, albeit complex, are descriptive enough to be understood by LLMs without background knowledge. Real-world enterprise databases break this assumption by naming fields with non human-readable names that are hard to interpret without business knowledge. When using LLMs naively on such enterprise databases, accuracy drops significantly. To close this performance gap, we study how knowledge graphs that explain the semantics of the schema can provide the missing knowledge. Our prototype DoubleAgent, a ReAct agent that can query databases and knowledge graphs *simultaneously* in a single reasoning loop, demonstrates that knowledge graphs can help restore accuracy on enterprise schemas from 10.9% to 43.6%.

### VLDB Workshop Reference Format:

Fares Elkholy, Timo Eckmann, Jan-Micha Bodensohn, Mulang' Onando, and Carsten Binnig. DoubleAgent: Towards Text-to-SQL on Enterprise Databases. VLDB 2026 Workshop: NOVAS Workshop.

## 1 INTRODUCTION

**Text-to-SQL has advanced significantly.** Text-to-SQL translates natural language requests into executable SQL queries, enabling users to query relational databases without writing SQL by hand. This line of work has seen rapid progress in recent years, driven by a succession of increasingly ambitious benchmarks, from Spider [11] to BIRD [6] to Spider 2.0 [5], each larger and more realistic than the last. Performance has climbed as well, with the strongest approaches surpassing 80% execution accuracy on BIRD and over 90% on the enterprise-scale Spider 2.0-Snow. While these existing benchmarks are almost saturated, the progress on them rests on the quiet assumption that the database schema, albeit complex in scale, is still understandable without background knowledge since its identifiers read like natural language. In enterprise systems, however, this is often not the case [1].

**Enterprise schemas not designed to be understood.** Enterprise databases often expose data through non human-readable (=cryptic) identifiers instead of descriptive attribute names. In SAP S/4HANA, for example, an attribute storing the net purchase order

value is named `POurgDocNetAmount`, and a delivery date becomes `BSTLDAT`. This is because schemas are often designed merely to hold the data for specific applications rather than to enable future understanding. Existing Text-to-SQL benchmarks do not fully reflect this reality. Spider, BIRD, and Spider2.0-Snow all use expressive English identifiers that map directly to natural language.

**Naively using LLMs does not work.** Faced with such cryptic schemas, simply prompting an LLM falls short. As the evaluation of state-of-the-art approaches on the enterprise benchmark BEAVER shows, performance is critically low [2], and we observe the same on a real SAP dataset. Importantly, we find that making the schema more enterprise-like by obscuring attribute names while leaving the questions and data untouched is enough to break Text-to-SQL agents like Spider-Agent, with accuracy dropping by roughly 30 percentage points.

**Introducing DoubleAgent.** While the schemas in production systems are often obscure, there typically exists a trove of business knowledge about these systems that supports their operation and maintenance. Many enterprises have recently begun to build and maintain knowledge graphs that encode this internal knowledge about business entities, domain concepts, and even semantic descriptions of individual attributes. For Text-to-SQL, such knowledge graphs can serve as the missing dictionary that maps cryptic physical identifiers to semantic business concepts in natural language. An agent that can query both sources thus gains access to both the *meaning* of the data (through the knowledge graph) as well as the *data itself* (through the database). Motivated by this insight, we propose the idea of a DoubleAgent that simultaneously issues SQL queries against the relational database and semantic queries against the business knowledge graph. Within a single ReAct [10] agent loop, DoubleAgent navigates the knowledge graph to *ground* its SQL against the actual meaning of the data.

**Contributions & outline.** In this paper, we present a prototype of DoubleAgent, a Text-to-SQL agent that combines knowledge graph exploration with SQL execution in a single agent loop (Section 2) and discuss our knowledge graph (Section 3). To support further research on the topic, we also present Spider 2.0-Cryptic, a benchmark based on Spider 2.0-Snow that incorporates the cryptic attribute names of real-world enterprise systems (Section 4). Finally, we evaluate DoubleAgent on Spider 2.0-Cryptic as well as in a real-world SAP S/4HANA setting with 516 OData questions, showing that it can restore accuracy from 10.9% to 43.6% on Spider 2.0-Cryptic and strict accuracy from 31.8% to 46.9% in a real-world setting directly on SAP data (Section 5).

---

*Equal contribution. Correspondence goes to timo.eckmann@tu-darmstadt.de
This work is licensed under the Creative Commons BY-NC-ND 4.0 International License. Visit https://creativecommons.org/licenses/by-nc-nd/4.0/ to view a copy of this license. For any use beyond those covered by this license, obtain permission by emailing info@vldb.org. Copyright is held by the owner/author(s). Publication rights licensed to the VLDB Endowment.
Proceedings of the VLDB Endowment. ISSN 2150-8097.

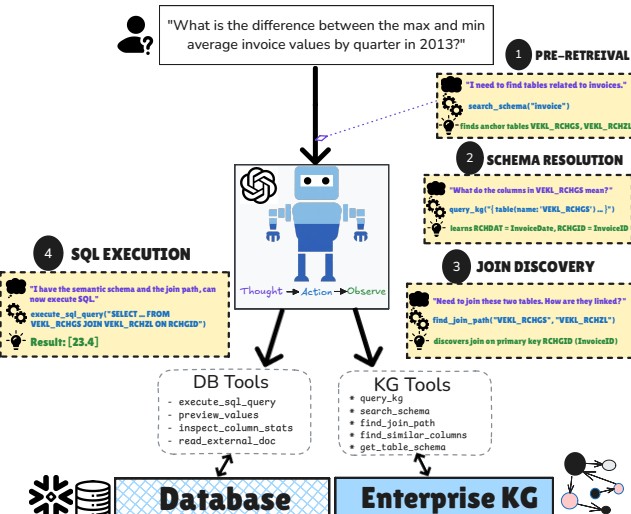

**Figure 1: To answer a user question, DoubleAgent first performs a vector-based schema pre-retrieval❶ to identify relevant anchor tables. It then enters a ReAct loop, alternating between querying the enterprise knowledge graph for semantic resolution❷❸ and executing SQL against the database,❹ before finally terminating via the `finish` tool.**

## 2 OUR SOLUTION: A DOUBLE AGENT

On enterprise databases, the hard part is not just transforming the query intent into SQL, but also working out what the columns mean. DoubleAgent therefore inverts the usual order. Before it writes a single query, it investigates the existing business knowledge graph to recover the semantic meaning of the obscure schema. This recovered meaning is then turned into SQL against the database.

**Finding the right parts of the schema.** Before the agent can reason at all, it faces a practical problem: A single database often stores tens of thousands of columns, far more than fit into the LLM's context window. Therefore, the agent must first find the tables and columns that the question actually needs. DoubleAgent thus starts with a schema retrieval step that runs before the agent loop starts (Figure 1). Following HybridRAG [8], it splits the question into short, schema-oriented phrases and embeds them with a sentence encoder [9]. It then uses vector search to find the table and column descriptions in the knowledge graph that are most relevant to the question, which become the anchor points for the agent.

**Deciding how much of the schema to show.** These anchors give a first idea about which columns matter, but not in what context. A column belongs to a table, and that table may have hundreds of other columns that the agent has to consider as well. How much of the surrounding schema to show to the agent depends on the database: For a small schema that fits into the LLM's context window, DoubleAgent shows it in full and simply marks the anchor columns. A large schema does not fit, and forcing it in would only bury the relevant columns among thousands of irrelevant columns. DoubleAgent therefore reduces the schema to ranked table summaries built around the anchors, and later expands summaries into full column lists on demand with a `get_table_schema` tool.

**Reasoning over the knowledge graph and database.** With the schema in context, either in full or as summaries the agent can expand, the main problem remains that the names are still cryptic, making it hard to map a question onto them. The agent must first resolve the meaning behind each name and then check its understanding against the data. Neither can be done in a single pass, since the agent often learns what a name means only after probing it. We therefore run a ReAct loop [10] that works both fronts at once. On the knowledge graph side, the agent looks up the business names and descriptions behind cryptic identifiers, loads relevant external documents into context, traces likely joins, and finds related columns. On the database side, it inspects sample values and executes SQL to test whether its mapping actually returns the right data. The agent alternates between both sides until it reaches a result in which it has confidence or exhausts a budget of 25 steps.

**Interacting with the knowledge graph and database.** DoubleAgent hinges on the tools that enable it to interact with the knowledge graph and database. The knowledge graph tools serve to recover meaning. At any step, the agent can traverse the knowledge graph with GraphQL [4] queries and thereby pull multiple properties such as business names, descriptions, and neighbors in a single call (query_kg). Furthermore, the agent can rerun the schema retrieval vector search to explore other parts of the knowledge graph (search_schema). Because enterprise schemas rarely declare explicit foreign-key relationships [5], it can also recover how two tables join by following LIKELY_FK edges in the knowledge graph (find_join_path), or surface columns that store the same thing under a different name by following SEM_SIMILAR edges (find _similar_columns). The database tools then put this recovered meaning to the test. Since a cryptic schema may also hide how its values are represented, the agent is hinted to first inspect the real formats and cardinalities (preview_values, inspect_column_stats) before it runs a query on the database (execute_sql_query). Importantly, the agent is instructed to treat an empty result as a signal rather than an answer. A repeatedly empty query therefore pushes the agent to inspect values and loosen filters until it finds the data. Finally, the agent must confirm its final result by calling `finish`.

## 3 OUR KNOWLEDGE GRAPH

As mentioned in the introduction, many enterprises have now begun to build knowledge graphs that encode their internal business knowledge about entities, domain concepts, and even semantic descriptions of individual attributes. For DoubleAgent, this knowledge graph provides the semantic context required to ground the SQL, serving as a dictionary that maps cryptic identifiers (e.g., BSTL_DAT) back to their business meaning. We model the knowledge graph as a *two-layer property graph*: a *cryptic layer*, which preserves the physical schema identifiers (e.g., LFRNG, AUFTR), and a *semantic layer* that holds their corresponding semantic names, descriptions, and data profiles (see Figure 2). DESCRIBES edges bind the two layers together, enabling the agent to reason in the semantic space while generating executable SQL against the cryptic schema.

Since many enterprise schemas do not provide explicit foreign-key/primary-key relationships and schema metadata is generally sparse, we must infer this missing context programmatically. We construct the semantic layer by first connecting it to the cryptic

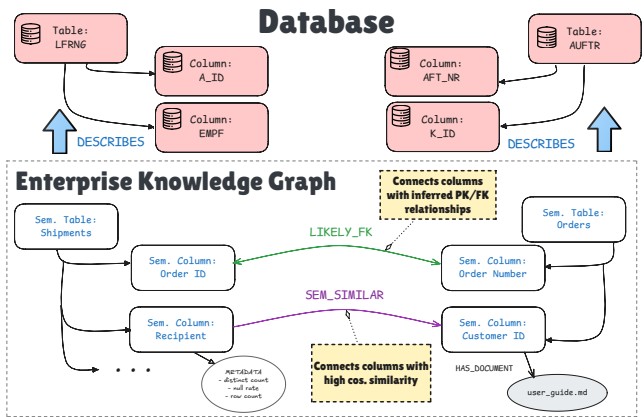

**Figure 2: Anatomy of an enterprise knowledge graph. Top: the *cryptic layer* of physical database identifiers. Bottom: the *semantic layer* of semantic schema nodes with descriptions, data profiles, metadata and external documents. Enrichment edges (LIKELY_FK & SEM_SIMILAR) capture likely joins and semantic similarity.**

layer via DESCRIBES edges. To compensate for the sparse metadata, we enrich the graph by generating descriptions for undescribed tables and columns using GPT-5-Mini. Next, we infer LIKELY_FK edges to recover join paths that are not explicitly enforced by the database. Specifically, we extract base entities from foreign-key suffixes (e.g., _id, _fk) and link them to corresponding primary keys, assigning a confidence score based on naming heuristics, data type equivalence, and value uniqueness.

Furthermore, we introduce SEM_SIMILAR edges to link pairs of columns whose underlying meaning (derived from semantic names, data types, and sample values) exhibits high cosine similarity. For instance, as shown in Figure 2, the semantic nodes "Recipient" and "Customer ID" reference the same underlying entity. Making this relationship explicit helps the agent avoid missing key columns during join operations. Finally, we embed every node in the knowledge graph using all-MiniLM-L6-v2 [9]. This allows the agent to retrieve relevant schema elements via vector search and inject a concise slice into its context, as detailed in Section 2.

## 4 A NEW TEXT-TO-SQL BENCHMARK

In this section, we describe the enterprise Text-to-SQL setting in more detail and introduce Spider 2.0-Cryptic, the first benchmark that pairs enterprise-scale schemas with cryptic SAP-style attribute names to enable a controlled analysis of the effect such cryptic names have on Text-to-SQL agents.

## 4.1 Why Are Existing Benchmarks Not Enough?

While Text-to-SQL benchmarks are slowly moving towards enterprise reality, no existing benchmark lets us measure the effect of enterprise-like cryptic schema names in isolation, with each benchmark falling short in a different way. Spider 1.0 uses tiny databases, with on average only 27.6 columns each [11]. BIRD is larger and noisier, yet its databases still average just 54 columns [6].

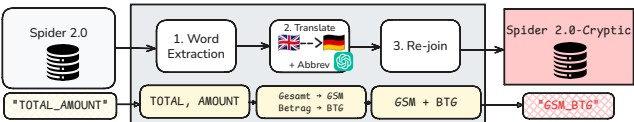

**Figure 3: To construct Spider 2.0-Cryptic, each Spider 2.0-Snow identifier is split into words, each word is translated into a German abbreviation, and the abbreviations are re-joined into a cryptic name.**

Spider 2.0 [5] is the first benchmark to reach enterprise scale, since its 213 public cloud databases average 744 columns and reach into the thousands in the largest cases. However, its identifiers remain descriptive English names. BEAVER instead samples from private enterprise warehouses, so its names are no longer descriptive [2]. Among these benchmarks, it is the most realistic; however, realism and isolation pull in opposite directions as BEAVER varies schema names, scale, data, and domain. Consequently, it becomes impossible to attribute any observed decline in accuracy to a single cause, such as the cryptic schema names. ADVETA provides both a clean as well as a cryptic version of the same schema by perturbing column names [7], but its perturbations are adversarial substitutions on small Spider tables, not the SAP-style abbreviations that span an entire database schema. Therefore, no existing benchmark provides both a clear and a cryptic version of the same enterprise-scale schema to isolate the effect of obfuscation.

## 4.2 Our New Benchmark: Spider 2.0-Cryptic

The goal of Spider 2.0-Cryptic is to enable the evaluation of enterprise-style schema obfuscation. To measure this dimension only, we start from Spider 2.0-Snow [5], a benchmark that is already enterprise-scale and readable, and rewrite only its schema names into cryptic SAP-style abbreviations while keeping the databases, questions, and data fixed. This produces a matching pair of schemas, identical in every aspect except their table and column names, making any drop in accuracy attributable to the SAP-style schema alone.

**Constructing Spider 2.0-Cryptic.** To make the cryptic names in Spider 2.0-Cryptic realistic, we studied a real S/4HANA OData dataset at SAP and imitated how its fields are abbreviated. To build Spider 2.0-Cryptic, we first select a 50-database subset of Spider 2.0-Snow [5], focusing on domains that are representative of enterprise data warehouses, such as government filings, healthcare, and e-commerce. Subsequently, as shown in Figure 3, we obfuscate only the schema names by (i) tokenizing every table and column name (e.g., using underscores and case changes as delimiters), which yields 5,288 unique English words, (ii) mapping each word to a German abbreviation of 2 to 6 characters with GPT-5-Mini, so that customer becomes KND and order becomes BSTL, and (iii) rejoining the abbreviations into the final cryptic identifier. We translate to German because SAP's physical names also derive from German business terms. Temporal suffixes such as _2022 are kept verbatim to preserve semantic partitioning. We then deploy each obfuscated schema as Snowflake views that alias the original names, so every query runs against the exact same data as Spider 2.0-Snow. Any difference in accuracy is therefore attributable to the schema names, and not to the data.

**Spider 2.0-Cryptic in numbers.** The resulting Spider 2.0-Cryptic dataset consists of clear and cryptic versions of the same enterprise-scale schemas, differing only in their names. In total, the benchmark spans 50 databases, 2,951 tables, and 259,482 columns, with 276 evaluation questions across 12 domains. The schema sizes are heavily skewed. While the median database holds 218 columns, the mean is around 5,200, and the largest database (FEC) reaches 485 tables and 71,819 columns. At this scale, an agent can no longer fit the full schema into its context. Finally, 56 questions (around 20%) are supported by external knowledge documents that contain proprietary domain knowledge and column descriptions.

## 5 INITIAL EVALUATION

Our initial experiments evaluate DoubleAgent on Spider 2.0-Cryptic and on a real SAP S/4HANA system. The results show that a knowledge graph lets the agent recover most of the accuracy lost due to cryptic schemas.

**Evaluation setup.** For Text-to-SQL, we use Spider 2.0-Cryptic with its 276 questions over a clear and a cryptic version of the schema. For the transfer study, we use 516 Text-to-OData questions over a real SAP S/4HANA system. On Spider 2.0-Cryptic, we report Execution Accuracy (EX), where a query is correct if its result table matches that of the gold query. On SAP S/4HANA, no live endpoint was available, so we instead report *strict accuracy*, where a generated OData query is correct only if it matches the gold query on all three of its parts, namely the target service and entity set, the filter predicates, and the selected fields. All agents use `GPT-5-Mini` with `high` thinking effort. We evaluate DoubleAgent with and without its knowledge graph, where disabling it removes the knowledge graph tools and the pre-retrieval, and compare it to Spider-Agent [5] as an external baseline.

**Exp. 1: Cryptic schemas break existing agents.** We first confirm that obfuscation alone is enough to break a state-of-the-art agent. Since the clear and the cryptic schemas differ only in their names, any drop must be caused by the obfuscation. Spider-Agent solves 53.6% of the clear questions but only 29.3% of the cryptic ones, a collapse of 24.3 percentage points (Figure 4, right). The abbreviated names give it no signal to link a question to the right tables and columns, and it does not recover from the resulting failures. This gap widens when external knowledge documents are missing. As shown on the left side of Figure 4, Spider-Agent then solves 41.6% of the clear questions but plummets to only 10.9% on the cryptic ones. By contrast, DoubleAgent is much more resilient and drops only from 51.5% to 43.6%.

**Exp. 2: The knowledge graph closes most of the gap.** Even without the knowledge graph, DoubleAgent already reaches 40.9% on cryptic schemas, well above Spider-Agent, since it retries failed queries to deduce names by trial and error. Adding the knowledge graph removes much of this guesswork and raises cryptic accuracy to 55.1%, cutting the obfuscation penalty from 13.5 to 5.3 percentage points. On clear schemas the gain is 6.0 percentage points, which confirms that the graph helps where the names are cryptic, not where they already carry meaning. An error analysis further shows that the knowledge graph nearly halves the number of errors due to schema-linking, from 83 to 44, so the dominant failure shifts from finding the right columns to writing correct SQL over them.

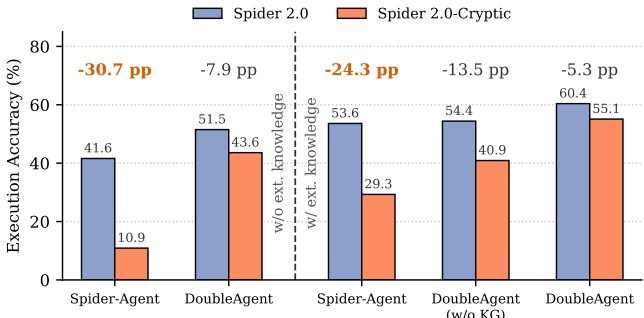

Figure 4: Execution Accuracy under Spider 2.0-Cryptic and Spider 2.0 ($n = 276$). Left: agents without external knowledge documents. Right: agents with access to external knowledge documents. The knowledge graph more than halves the accuracy drop that obfuscation causes.

Table 1: SAP results ($n = 516$). The knowledge graph improves strict accuracy & lowers cost, at the price of higher latency.

|  | Strict accuracy | Cost ($) | Runtime (s) |
|---|---|---|---|
| Baseline | 31.8% | 4.92 | **0.81** |
| With KG | **46.9%** | **4.22** | 2.45 |

**Exp. 3: The gain transfers to SAP S/4HANA.** Finally, the same approach transfers beyond SQL to a real SAP S/4HANA system. On 516 OData questions across 11 business domains, the knowledge graph raises strict accuracy from 31.8% to 46.9% (Table 1). The gain is larger than on SQL, at ~15 against ~11 percentage points, since here the knowledge graph is crucial to route each question to the correct service among thousands. In addition, it slightly reduces the cost, from $4.92 to $4.22, since the agent wastes fewer steps in the exploration, although the knowledge graph-variant adds latency as the agent's SPARQL needs to be executed on the knowledge graph.

## 6 FUTURE DIRECTIONS

Spider 2.0-Cryptic and DoubleAgent are first steps towards Text-to-SQL on enterprise databases, and leave clear directions open.

**Completing the knowledge graph.** Our knowledge graph is built automatically and is therefore incomplete. We infer keys from name patterns, generate descriptions, and add semantic edges, all of which are approximate. Since real enterprise knowledge graphs are also often incomplete, a natural next step is automatic knowledge graph completion, where missing keys, descriptions, and relationships are recovered more reliably than our heuristics allow.

**Routing to the right data source.** A preliminary error analysis on the SAP setting points to a sharper problem. The largest source of errors is not writing an incorrect query but selecting the wrong data source, since the agent must first route a question to the correct OData service among thousands before it can answer it. This is no longer a single-database problem and instead requires finding and combining the right sources, which is exactly the setting of our work on autonomous data agent collaboration, where a coordinator routes a question to the agents that can answer it [3]. We therefore intend to extend DoubleAgent in this direction.

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
