# OpenReview forum: "DoubleAgent: Towards Text-to-SQL on Enterprise Databases"
_VLDB.org/2026/Workshop/NOVAS — NOVAS 2026_

### Official Review · Reviewer_cGU9 · 2026-06-29

**Confidence:** 3

**Improvement Opportunities:**

O1. Given that the schema is deeply cryptic, it is unclear how to bootstrap the knowledge graph construction. How can a lightweight LLM reliably deduce the semantic meaning of a cryptic schema? What external metadata did the authors feed into the LLM during the knowledge graph construction?

O2. In Section 2, the authors state that “the agent is instructed to treat an empty result as a signal rather than an answer”. What if the empty result is the correct answer (which might be common in practice)?

O3. The authors use suffix matching to infer foreign keys (Section 3). Relying on this heuristic seems somewhat contradictory to the paper's premise. In deeply cryptic schemas, foreign keys are unlikely to use such clean English suffixes.

**Minor Comments:**

M1. The text in Figure 1 is way too small to parse.

M2. Typo in Figure 1: “PRE-RETREIVAL” → “PRE-RETRIEVAL”.

**Short Summary:**

This paper tackles the problem in Text-to-SQL that highly cryptic, abbreviation-heavy schemas cannot be recognized semantically by the LLM. The paper, therefore, presents DoubleAgent that simultaneously queries an Enterprise Knowledge Graph and the physical database within a single reasoning loop. To evaluate this, the authors introduce Spider 2.0-Cryptic, a novel benchmark that isolates the impact of schema obfuscation by translating English schema identifiers into SAP-style German abbreviations. The evaluation shows that DoubleAgent recovers a significant portion of the accuracy lost to cryptic names on the benchmark.

**Strong Points:**

S1. The paper has a strong practical motivation by addressing the messy reality of enterprise database schemas.

S2. The Spider 2.0-Cryptic benchmark design is clever. It effectively isolates “schema readability” to conduct controlled ablation.

S3. The solution is evaluated on a real-world production-like SAP S/4HANA dataset.

---

### Official Review · Reviewer_N1rt · 2026-06-30

**Confidence:** 4

**Improvement Opportunities:**

I1) What happens if a cryptic attribute/table name, or an FK relationship, is not recorded (or discovered with additional tools) in the knowledge graph? Does DoubleAgent return an error, or does it still attempt to generate the corresponding SQL query? Moreover, real user queries may return empty results, especially when users are exploring the data. How does your retry criterion work in this case? Does the agent keep trying until the maximum number of iterations is reached?

I2) How do you measure whether the SQL query returns the "right data" during the agentic loop? Do you use any heuristics? While it is easy to detect errors when the SQL query is expected to return a different type from the expected, it can be much more difficult when the query returns the same type of result but with a different, yet plausible, value.

**Minor Comments:**

M1) The font size in the yellow boxes in Figure 1 could be increased.

**Short Summary:**

The paper presents DoubleAgent, a Text-to-SQL agent designed to handle real enterprise NL2SQL queries, where the target schema may contain cryptic attribute/table names. To support the evaluation of DoubleAgent, the authors also introduce a new benchmark dataset generated from Spider 2.0, in which schema names are replaced with cryptic ones while the underlying data remains unchanged. The evaluation is conducted both on the generated dataset and on a real enterprise dataset. The proposed approach shows promising results, although it does not completely solve the problem. However, it identifies relevant future research directions for closing the existing gap.

**Strong Points:**

S1) The paper tackles an interesting and real-world problem and shows that DoubleAgent can mitigate the issue of obfuscated names in enterprise databases.

S2) The paper introduces a new benchmark dataset that focuses specifically on the problem of obfuscated names. This could be a useful resource for further research in this direction.

S3) The paper is well written and easy to follow.

---

### Official Review · Reviewer_4hJ2 · 2026-07-08

**Confidence:** 5

**Improvement Opportunities:**

1. It is not very clear whether using the knowledge graph for the Spider 2.0 cryptic version actually cancels out obfuscation (whether the KG restores connections of cryptic german abbreviations to the original english column headers). The authors would need to clarify how the KG looks like in the Spider 2.0 cryptic benchmark they constructed.
2. The authors need to elaborate on why Spider-Agent performs much worse than DoubleAgent in the case where both have access to external knowledge but not KGs (right part of Figure 4): when stripping KGs from both, I would expect them to have identical or similar performance. What makes DoubleAgent performing much better in this case?
3. Apart from accuracy, it would be insightful to also summarize the traces of the agents in each case, meaning how do they operate (which tool calls they mostly make) depending on the information they have available.

**Minor Comments:**

(Points for discussing future directions/evaluations)
1. How does an agent perform when there are missing/erroneous relationships in the KG? Should the agent also reason on the KG information?
2. Cryptic column names cave a different effect on numerical/categorical/textual data, since for some of these cases the agent can extract semantics from data instances and others not (and context too). It would be beneficial in a future version of the method to breakdown evaluation on the data nature that columns store and how this affects accuracy.

**Short Summary:**

The paper introduces DoubleAgent an agent-based text-to-sql solution that leverages knowledge graphs, besides access to external documents and actual metada/data in relational databases; the authors' goal is mainly to showcase the importance of utilizing knowledge graphs in order to improve performance on realistic enterprise-db scenarios. To evaluate their method, the authors create a Spider-2.0 cryptic version and results justify the importance of building agentic methods that know how to leverage semantic layers (like knowledge graphs) for Text-to-SQL.

**Strong Points:**

1. The motivation for a method that can leverage information found in knowledge graphs is very clearly stated and well-justified in the introduction.
2. The setting that the authors discuss is very well aligned with current trends in building semantic layers upon data collections for the agents to utilize.
3. The authors have created a cryptic-version of Spider 2.0 to showcase their results which justify their earlier claims.

---

### Decision · Program_Chairs · 2026-07-16

**Decision:**

Accept

**Comment:**

DoubleAgent addresses a practical Text-to-SQL challenge by combining database access with enterprise knowledge graphs to handle cryptic schemas. The paper is well motivated, introduces a useful Spider 2.0-Cryptic benchmark, and demonstrates promising results on both controlled and real enterprise settings. Its focus on semantic layers is highly relevant to agentic data systems, and we hope it sparks interesting discussion at the workshop.